# Sex-specific association of visceral adiposity index with renal dysfunction in chinese type 2 diabetes: A cross-sectional study

Songfang Liu[1,2], Louyan Ma[3], Yu Niu[2], Ranran Ma[4], Ting Qi[2], Bingyin Shi🄳[1]*

**1** Department of Endocrinology, The First Affiliated Hospital of Xi'an Jiaotong University, Xi'an, Shaanxi, China, **2** Department of Endocrinology, Ninth Hospital of Xi'an, Xi'an, Shaanxi, China, **3** Department of General Practice Medicine, Ninth Hospital of Xi'an, Xi'an, Shaanxi, China, **4** Department of Neurology, Ninth Hospital of Xi'an, Xi'an, Shaanxi, China

* shibingy@126.com

## Abstract

### Background

Diabetic nephropathy is a serious microvascular complication of type 2 diabetes mellitus, yet the sex-specific relationship between visceral adiposity and renal dysfunction remains insufficiently explored in Chinese populations.

### Objective

To investigate the association between visceral adiposity indices and renal dysfunction in Chinese type 2 diabetes mellitus patients, emphasizing this population's unique risk profile and the clinical utility of composite adiposity measures.

### Methods

This cross-sectional study involved 1,335 type 2 diabetes mellitus patients. The primary outcome variable was renal dysfunction, defined as an estimated glomerular filtration rate (eGFR) < 60 mL/min/1.73 m². Thirteen visceral adiposity indices were evaluated through correlation analysis, overall population regression modeling, and sex-stratified approaches. Multiple regression models were progressively adjusted for demographic characteristics, lifestyle factors, clinical parameters, and medications to assess independent associations with estimated glomerular filtration rate.

### Results

The study population (median age: 54 years; 67.04% male) demonstrated distinct characteristics between estimated glomerular filtration rate groups. The estimated glomerular filtration rate decline group exhibited significantly higher age, blood pressure, and fasting glucose compared to the normal group (all p < 0.05). Despite similar body mass index levels, multiple visceral adiposity indices (waist-to-hip

**Data availability statement:** All relevant data are within the paper and its Supporting information files.

**Funding:** The author(s) received no specific funding for this work.

**Competing interests:** The authors have declared that no competing interests exist.

**Abbreviations:** SBP, Systolic Blood Pressure; DBP, Diastolic Blood Pressure; HbA1c, Glycated Hemoglobin; FPG, Fasting Plasma Glucose; TC, Total Cholesterol; TG, Triglycerides; HDL, High-Density Lipoprotein; LDL, Low-Density Lipoprotein; eGFR, Estimated Glomerular Filtration Rate; ARB Use, Angiotensin Receptor Blockers Use; WC, Waist Circumference; BMI, Body Mass Index; WHR, Waist-to-Hip Ratio; WHtR, Waist-to-Height Ratio; LAP, Lipid Accumulation Product; CVAI, Chinese Visceral Adiposity Index; VAI, Visceral Adiposity Index; ABSI, A Body Shape Index; BRI, Body Roundness Index; RFM, Relative Fat Mass; FLI, Fatty Liver Index; METS-VF, Metabolic Score for Visceral Fat; METS-IR, Metabolic Score for Insulin Resistance.

ratio, lipid accumulation product, Chinese visceral adiposity index, visceral adiposity index, body roundness index, relative fat mass, and metabolic score for visceral fat) were significantly elevated in the eGFR Decline Group. Comprehensive correlation analysis revealed substantial attenuation after confounder adjustment, with only visceral adiposity index maintaining significance in the total population (adjusted r = −0.075, p = 0.007). Multivariate regression confirmed visceral adiposity index as the most robust predictor (β = −1.63, 95% CI: −2.75 to −0.50, p = 0.0048). Sex-stratified analyses revealed profound differences: visceral adiposity index demonstrated independent predictive value exclusively in males (β = −2.41, 95% CI: −3.84 to −0.98, p = 0.001), while all female associations became non-significant after full adjustment.

## Conclusion

Visceral adiposity index showed strong and consistent associations with renal dysfunction specifically in Chinese male T2DM patients, while female associations were primarily mediated through confounding pathways. These findings highlight critical sex-specific differences and support the integration of composite visceral adiposity indices into routine clinical assessment for male diabetic patients, emphasizing the necessity of population-specific and gender-tailored approaches in renal risk stratification.

## Introduction

Type 2 diabetes mellitus (T2DM) is a major global public health challenge, with the International Diabetes Federation reporting 537 million affected individuals worldwide, projected to reach 783 million by 2045 [1]. In China, adult diabetes prevalence has reached 10.9%, affecting approximately 140 million individuals [2]. Diabetic nephropathy, a major microvascular complication affecting 20–40% of T2DM patients, significantly increases morbidity and cardiovascular mortality risk [3,4]. Recent multicenter data indicate that chronic kidney disease prevalence among Chinese T2DM patients is approximately 31% [5], highlighting the urgent need for early identification and management strategies.

While conventional anthropometric parameters such as body mass index (BMI) and waist-to-hip ratio (WHR) offer practical utility, they inadequately reflect visceral fat distribution [6]. Novel anthropometric indices, including the fatty liver index (FLI), visceral adiposity index (VAI), and Chinese visceral adiposity index (CVAI), integrate anthropometric and biochemical parameters to provide enhanced assessment of visceral adiposity [7]. These metrics have demonstrated significant associations with chronic disease risk factors, including cardiovascular disease and metabolic syndrome [8,9].

However, the relationship between visceral adiposity and renal function impairment remains controversial in current literature. While some studies have identified

negative correlations between VAI and declining eGFR, others have failed to establish significant associations [10]. These inconsistencies may arise from differences in study populations and methodologies. Notably, there is a significant gap in systematic comparisons of various visceral adiposity indices for predicting renal dysfunction, particularly in Chinese T2DM populations who exhibit distinct fat distribution patterns and metabolic risks compared to Western populations.

This cross-sectional study systematically evaluates the association between thirteen visceral adiposity indices and renal dysfunction in Chinese T2DM patients, comparing their predictive capabilities. This represents the first comprehensive assessment of these indices in this specific population, utilizing multivariate statistical methods while controlling for potential confounding factors. The findings aim to optimize screening tools for renal dysfunction risk and provide evidence-based guidance for early prevention strategies in Chinese T2DM patients.

## Methods

### Study population

This cross-sectional study was conducted at the Ninth Hospital of Xi'an, China, from July 1, 2020, to June 30, 2022. A total of 1,335 adult patients with confirmed type 2 diabetes mellitus (T2DM) were included. For the purposes of this retrospective analysis, the data from electronic medical records and laboratory systems were accessed between August 1, 2022, and September 30, 2022. Eligibility criteria were as follows: (1) age ≥ 18 years; (2) diagnosis consistent with the World Health Organization's 1999 criteria for diabetes, defined by fasting plasma glucose (FPG) ≥7.0 mmol/L, 2-hour glucose ≥11.1 mmol/L during an oral glucose tolerance test (OGTT), or glycated hemoglobin (HbA1c) ≥6.5%; and (3) availability of complete data for exposure and outcome variables. Exclusion criteria included: (1) incomplete data on visceral fat accumulation or renal function; (2) presence of severe acute or chronic conditions (e.g., malignancy, heart failure); or (3) pregnancy. Data were extracted from inpatient medical records and laboratory results. Throughout the data collection and analysis period, the research team had access only to anonymized, de-identified datasets. All personal identifiers (such as name, ID number, contact information) were removed prior to data analysis, ensuring that authors could not identify individual participants. To ensure accuracy and consistency, trained personnel used standardized forms for data collection and implemented a cross-verification process.

### Sample size calculation

Sample size was calculated using PASS 15.0 software (NCSS, LLC, Kaysville, Utah, USA) based on preliminary data suggesting a moderate correlation ($r = 0.15$) between visceral adiposity indices and eGFR in diabetic populations. Using a two-sided significance level of $\alpha = 0.05$ and power of 80%, the minimum required sample size was 692 participants for detecting this effect size. To accommodate comprehensive sex-stratified analyses and account for potential missing data exclusions, we aimed for approximately 1,300 participants. Ultimately, 1,335 eligible participants were included in the final analysis, exceeding the minimum requirement.

### Variables

Thirteen anthropometric and biochemical indices were calculated to comprehensively assess visceral adiposity: Body Mass Index (BMI), Waist Circumference (WC), Waist-to-Hip Ratio (WHR), Waist-to-Height Ratio (WHtR), Lipid Accumulation Product (LAP), Visceral Adiposity Index (VAI), Chinese Visceral Adiposity Index (CVAI), A Body Shape Index (ABSI), Body Roundness Index (BRI), Relative Fat Mass (RFM), Weight-adjusted Waist Index (WWI), Fatty Liver Index (FLI), and Metabolic Score for Visceral Fat (METS_VF). (see Table 1 for detailed computational methods.) All anthropometric measurements were performed during the first 24 hours post-admission under standardized conditions using calibrated equipment by trained personnel.

**Table 1. Anthropometric indices assessed in the study.**

| BMI | All | weight (kg)/ (height [cm]/100)² |
|---|---|---|
| WHR | All | WC[cm]/ Hip Circumference[cm] |
| WHtR | All | WC [cm]/ Height [cm] |
| LAP | Male | (WC[cm] − 65) × TG[mmol/L] |
| | Famle | (WC[cm] − 58) × TG[mmol/L] |
| CVAI | Male | −267.93 + 0.68 × Age + 0.03 × BMI + 4.00 × WC[cm] |
| | Famle | −187.32 + 1.71 × Age + 4.23 × BMI + 1.12 × WC[cm] |
| VAI | Male | (WC[cm]/39.68 + (1.88 × BMI)) × (TG[mmol/L]/1.03) × (1.31/HDL-C [mmol/L]) |
| | Famle | (WC[cm]/36.58 + (1.89 × BMI)) × (TG[mmol/L]/0.81) × (1.52/HDL-C [mmol/L]) |
| ABSI | All | (WC[cm]/100)/ (BMI²ᐟ³ × (Height[cm]/100)½) |
| BRI | All | 364.2 - 365.5 × √(1 − (((WC[cm]/100)/2π)²/ ((Height[cm]/100)/2)²)) |
| RFM | Male | 64 − (20 × Height[cm]/WC[cm]) |
| | Famle | 76 − (20 × Height[cm]/WC[cm]) |
| FLI | All | (1/ (1 + e^(−0.953 × ln(TG[mg/dL]) − 0.139 × BMI − 0.718 × ln(GGT[U/L]) − 0.053 × WC[cm] + 15.745))) × 100 |
| METS-VF | All | 39.61 + (1.04 × BMI) + (0.88 × WC[cm]) − (0.81 × Age) |
| METS-IR | All | Ln((2 × TG[mg/dL] × Glu[mg/dL] × BMI)/ (HDL-C[mg/dL] × Ln(2))) |

The primary outcome variable of this study was renal function. It was defined as an estimated glomerular filtration rate (eGFR) < 60 ml/min/1.73m², which corresponds to the clinical cutoff for moderate chronic kidney disease(CKD stage 3) according to the KDIGO guidelines. eGFR was calculated using the Modification of Diet in Renal Disease (MDRD) equation [11]. Accordingly, participants were classified into two groups for analysis: the eGFR Normal Group (eGFR ≥ 60 mL/min/1.73 m²) and eGFR Decline Group (eGFR < 60 mL/min/1.73 m²).

Covariates were selected through an extensive literature review and expert consultation, covering five critical domains: demographic characteristics (age and sex), lifestyle factors (smoking and alcohol consumption), clinical measurements (blood pressure and diabetes duration), laboratory parameters (fasting plasma glucose and cholesterol levels), and therapeutic interventions (use of Angiotensin II Receptor Blockers).

## Statistical analysis

This study employed a comprehensive three-phase statistical approach: (1) correlation analysis, (2) overall population regression modeling, and (3) sex-stratified analyses. Descriptive statistics utilized median (interquartile range) for continuous variables and percentages for categorical variables, with group comparisons using Mann-Whitney U and chi-square tests. Correlation analyses examined both unadjusted and adjusted associations between 13 visceral adiposity indices and eGFR, controlling for confounders including age, sex, diabetes duration, HbA1c, blood pressure, and medication use. Multiple linear regression employed three progressive models with standardized adiposity indices: Model 1 (unadjusted), Model 2 (age and sex adjusted), and Model 3 (fully adjusted for demographic, lifestyle, clinical, and therapeutic variables). Comprehensive sex-stratified analyses included separate correlation and regression modeling to identify gender-specific associations. All analyses used R version 4.3.2, with statistical significance at P < 0.05. We adopted a rigorous approach to data completeness. Participants with missing data on primary exposure variables (anthropometric measurements) or outcome variables (renal function indicators) were strictly excluded from the study. For potential confounding covariates where missing data was minimal (<5%), multiple imputation was employed to maximize data utilization and reduce bias.

### Ethic statement

The Medical Ethics Committee of the hospital approved this research protocol (reference: 2021−38), granting a waiver of informed consent due to the retrospective nature of data collection and difficulties in contacting some patients. This decision complies with local regulations and the Declaration of Helsinki. The research team followed medical ethical guidelines meticulously, ensuring patient confidentiality and data security. Collected information was used solely for academic research purposes, with strict protocols against any other use.

## Results

### Comprehensive characterization of a type 2 diabetes cohort

A total of 1,335 patients with type 2 diabetes mellitus (T2DM) were included in this study. The median age was 54 years (IQR: 47–62), with 67.04% of the cohort being male. The median duration of diabetes was 8 years (IQR: 3–13). Glycemic control indicators included an HbA1c of 8.5% (IQR: 7.2–10.1) and a fasting plasma glucose (FPG) level of 8.1 mmol/L (IQR: 6.6–10.4). Blood pressure measurements showed median systolic and diastolic values of 130 mmHg (IQR: 120–140) and 80 mmHg (IQR: 70–86), respectively (Table 2). Table 2 also presents other baseline clinical parameters, including lifestyle factors, history of angiotensin receptor blocker (ARB) therapy, serum biochemical parameters, and anthropometric measurements for adiposity assessment among the study cohort.The comprehensive baseline characteristics underscore the complex metabolic interactions in this patient population.(Table 2) Given the Chinese-specific context (mean BMI lower than many Western cohorts), these findings may underscore unique body composition trends found in East Asian populations.

### Comparative analysis of eGFR decline group characteristics

As shown in Table 3,compared with the eGFR Normal Group, patients in the eGFR Decline Group demonstrated significantly distinct demographic and clinical characteristics. The eGFR Decline Group was characterized by advanced age (62 [54, 70] years vs. 53 [46, 60] years, p<0.001) and exhibited markedly elevated blood pressure parameters, with higher systolic (140 [130, 150] mmHg vs. 126 [120, 140] mmHg, p<0.001) and diastolic blood pressure (80 [70, 85] mmHg vs. 80 [75, 90] mmHg, p=0.0174). Notably, the eGFR Decline Group presented with impaired metabolic profiles, evidenced by elevated fasting plasma glucose (8.35 [6.91, 11.7] vs. 8.0 [6.5, 10.2] mmol/L, p=0.009) and significantly altered lipid parameters, including increased TC and TG levels (all p<0.05).

Although there were no significant differences in BMI between the two groups, several visceral fat metabolism indices (such as WHR, LAP, CVAI, VAI, BRI, RFM, and METSVF) were significantly higher in the eGFR Decline Group. This indicates that, while BMI is a commonly used tool for assessing obesity, it fails to adequately reflect the complexities of visceral fat and metabolic abnormalities. This further emphasizes the potential association between metabolic dysregulation of visceral fat and renal deterioration, suggesting that a more comprehensive set of indices should be considered when evaluating the impact of obesity on health. These findings highlight the potential role of adipose tissue as an active endocrine organ contributing to systemic inflammation and organ damage. The significant differences in sex distribution, smoking status, alcohol consumption and angiotensin receptor blocker (ARB) utilization further emphasize the multifactorial nature of renal function decline. (Table 3)

### Comprehensive statistical analysis of anthropometric-eGFR associations

**Initial correlation analysis.** Preliminary correlation analyses across the total population of 1,335 participants revealed significant negative associations between multiple anthropometric indices and eGFR (Fig 1). In unadjusted analysis (Fig 1A), relative fat mass (RFM) demonstrated the strongest correlation (r=−0.159, p<0.001), followed by Chinese visceral adiposity index (CVAI, r=−0.117, p<0.001) and a body shape index (ABSI, r=−0.080, p=0.003). Additional significant correlations were observed for body roundness index (BRI, r=−0.064, p=0.02) and waist-to-height ratio (WHtR, r=−0.063, p=0.021).

**Table 2. Baseline Demographic, Clinical, and Anthropometric Characteristics of Patients with Type 2 Diabetes.**

| Variable | Result |
|---|---|
| number | 1335 |
| Age,years | 54 (47, 62) |
| Sex | |
| Male,% | 67.04 |
| Femal,% | 32.96 |
| diabetes duration,years | 8 (3, 13) |
| SBP,mmHg | 130 (120, 140) |
| DBP,mmHg | 80 (70, 86) |
| HBA1C,% | 8.5 (7.2, 10.1) |
| FPG,mmol/L | 8.1 (6.555, 10.4) |
| TC,mmol/L | 4.09 (3.48, 4.73) |
| TG,mmol/L | 1.56 (1.05, 2.34) |
| HDL-c,mmol/L | 0.94 (0.8, 1.13) |
| LDL-c,mmol/L | 2.33 (1.84, 2.845) |
| eGFR,ml/min/1.73m$^2$ | 78.48 (66.48, 88.89) |
| Smoking history,% | |
| never | 54.53 |
| current | 35.58 |
| former | 9.89 |
| Alcohol history,%,% | |
| never | 73.03 |
| current | 23 |
| former | 3.97 |
| ARB use,% | 21.65: % |
| WC,cm | 90 (85, 97) |
| BMI,kg/m2 | 25.5 (23.5, 27.7) |
| WHR | 0.93 (0.89, 0.97) |
| WHtR | 0.54 (0.51, 0.58) |
| LAP | 42.55 (25.2, 71.28) |
| CVAI | 129.90(97.61, 158.25) |
| VAI | 2.41 (1.49, 4.04) |
| ABSI | 0.08(0.077, 0.084) |
| BRI | 4.21(3.47, 5.02) |
| RFM | 29 (25.78, 36.5) |
| FLI | 42.79 (21.11, 65.91) |
| METS-VF | 6.42 (6.11, 6.69) |
| METS-IR | 9.91 (9.05, 10.78) |

Results are presented as numerical values, proportions (%), or median (Q1–Q3).

However, after adjustment for potential confounders (Fig 1B), only visceral adiposity index (VAI) maintained statistical significance (adjusted r=−0.075, p=0.007), indicating substantial mediation by confounding variables.

 **Overall population multivariate regression analysis.** Building upon the correlation findings, multivariate regression analyses through three progressive models revealed significant associations between various anthropometric and

**Table 3. Demographic, Clinical, and Metabolic Parameters in Patients with Normal and Declined eGFR.**

| Variable | eGFR Normal Group (n = 1161) | eGFR Decline Group (n = 174) | P_Value |
|---|---|---|---|
| Age,years | 53 (46, 60) | 62 (54, 70) | <0.001 |
| Sex,male/female | 69.16%/30.84% | 52.87%/47.13% | <0.001 |
| diabetes duration,years | 7 (2, 12) | 12 (8, 18) | <0.001 |
| Smoking history (never/current/former) | 53.23%/37.47%/9.3% | 63.22%/22.99%/13.79% | <0.001 |
| Alcohol history (never/current/former) | 71.9%/24.72%/3.36% | 80.46%/11.49%/8.05% | <0.001 |
| ARB, users/non-users | 81.57%/18.43% | 56.9%/43.1% | <0.001 |
| SBP, mmHg | 126 (120, 140) | 140 (130, 150) | <0.001 |
| DBP, mmHg | 80 (70, 85) | 80 (75, 90) | 0.017 |
| HbA1c, % | 8.5 (7.3, 10.1) | 8.35 (7.02, 9.95) | 0.459 |
| FPG, mmol/L | 8 (6.5, 10.2) | 8.35 (6.91, 11.7) | 0.009 |
| TC,mmol/L | 4.06 (3.47, 4.69) | 4.23 (3.55, 4.76) | 0.020 |
| TG,mmol/L | 1.54 (1.05, 2.31) | 1.72 (1.15, 2.51) | 0.027 |
| HDL,mmol/L | 0.94 (0.8, 1.13) | 0.96 (0.78, 1.18) | 0.477 |
| LDL,mmol/L | 2.32 (1.85, 2.84) | 2.36 (1.83, 3.07) | 0.348 |
| eGFR,ml/min/1.73m$^2$ | 80.82 (71.78, 90.56) | 53.11 (48.24, 57.38) | <0.001 |
| WC,cm | 90 (84, 96) | 91.5 (85, 97) | 0.455 |
| BMI,kg/m$^2$ | 25.5 (23.5, 27.7) | 25.4 (23.62, 27.85) | 0.787 |
| WHR | 0.92 (0.89, 0.97) | 0.94 (0.89, 0.98) | 0.027 |
| LAP | 41.65 (25, 68.82) | 51.73 (28.05, 78.77) | 0.016 |
| CVAI | 126.15 (94.95, 155.96) | 143.24 (117.74, 167.58) | <0.001 |
| VAI | 2.37 (1.48, 3.95) | 2.84 (1.66, 4.75) | 0.011 |
| BRI | 4.18 (3.47, 4.99) | 4.52 (3.59, 5.36) | 0.006 |
| RFM | 28.64 (25.55, 35.51) | 31.28 (27.13, 40.50) | <0.001 |
| ABSI | 0.08062(0.07765, 0.08369) | 0.08133(0.07790, 0.08453) | 0.143 |
| FLI | 41.11 (20.89, 65.64) | 46.85 (25.06, 68.50) | 0.246 |
| METS-VF | 6.40 (6.09, 6.68) | 6.57 (6.23, 6.79) | <0.001 |
| METS-IR | 9.89 (9.04, 10.77) | 910.01(9.03, 10.90) | 0.522 |

Results are presented as numerical values, proportions (%), or median (Q1–Q3).

metabolic indices and eGFR outcomes (Fig 2). In the initial model, CVAI demonstrated the strongest negative association (β = −2.14, 95% CI: −3.04 to −1.24, p < 0.001, R² = 0.015), followed by comparable effects from METS_VF (β = −2.16, 95% CI: −3.05 to −1.25, p < 0.001) and WWI (β = −2.03, 95% CI: −2.93 to −1.13, p = <0.001). After adjusting for confounding variables in Model 2, LAP (β = −1.18, 95% CI: −2.00 to −0.35, p = 0.005) and FLI (β = −1.43, 95% CI: −2.27 to −0.60, p < 0.001) emerged as significant predictors, with improved model performance (R² ≈ 0.19). The fully adjusted Model 3 revealed VAI as the most robust predictor (β = −1.63, 95% CI: −2.75 to −0.50, p = 0.0048, R² = 0.25), while other adiposity distribution indices showed attenuated effects.

**Sex-stratified correlation and regression analysis.** Sex-stratified analysis revealed marked differences in anthropometric-eGFR associations (Fig 3). In male participants (n = 895), unadjusted correlations were generally weak (Fig 3A), with only METS_VF demonstrating significance (r = −0.130, p < 0.001). After adjustment (Fig 3B), VAI emerged as the sole robust predictor (adjusted r = −0.103, p < 0.01). Regression analysis confirmed VAI's prominence in males

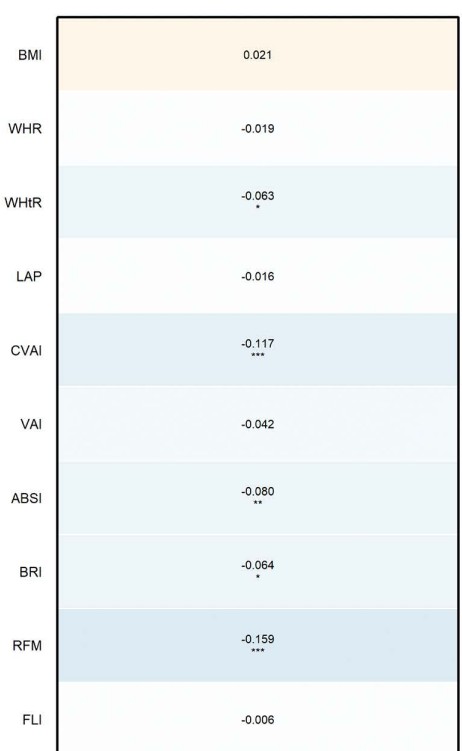

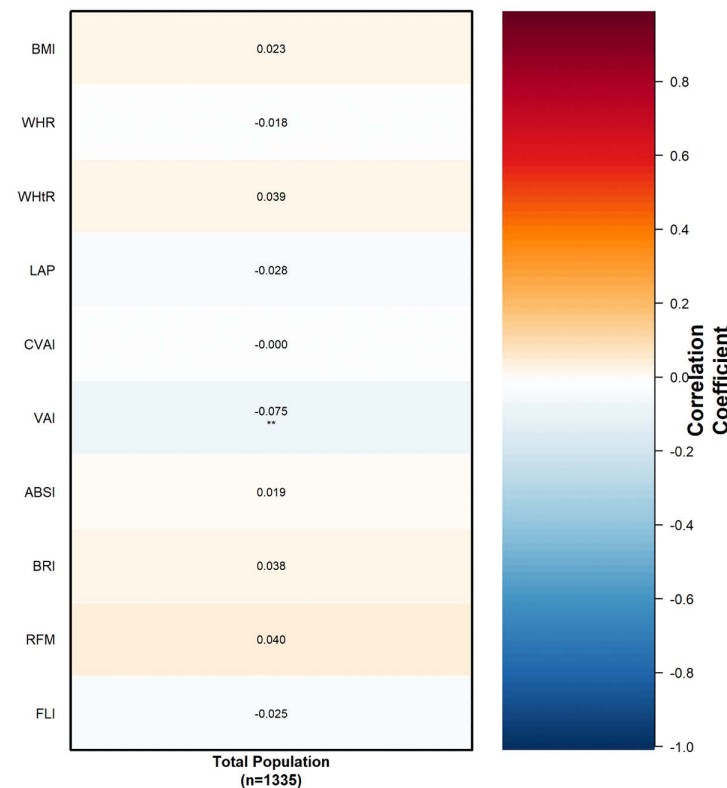

**Fig 1. Correlations between visceral adiposity indices and estimated glomerular filtration rate in the total population.** (A) Unadjusted Pearson correlation coefficients between 13 visceral adiposity indices and estimated glomerular filtration rate (GFR.MORD). (B) Partial correlation coefficients adjusted for potential confounding variables (AGE, SMOKING, WINE, BP, BP1, TIME, FPG, TC, HDL, LDL, ARB, SEX). Heat map colors represent the strength and direction of correlations, with blue indicating negative correlations and red indicating positive correlations. Values in cells represent correlation coefficients with statistical significance indicated by asterisks: *** P<0.001, ** P<0.01, * P<0.05. Total sample size: n=1335.

(Fig 4), particularly in the fully adjusted model (β=−2.41, 95% CI: −3.84 to −0.98, p=0.001), while WWI showed a significant positive association (β=1.20, 95% CI: 0.16 to 2.23, p=0.024).

In contrast, female participants (n=440) exhibited extensive significant negative correlations in unadjusted analysis (Fig 3A), with CVAI demonstrating the strongest association (r=−0.274, p<0.001), representing a medium-to-large effect size, followed by METS_VF (r=−0.216, p<0.001). Other significant correlations included ABSI (r=−0.145, p<0.01), LAP (r=−0.127, p<0.01), RFM (r=−0.121, p<0.05), WHtR (r=−0.118, p<0.05), BRI (r=−0.114, p<0.05), FLI (r=−0.098, p<0.05), and waist circumference (r=−0.095, p<0.05). After adjustment (Fig 3B), all female associations became non-significant. In regression analysis (Fig 5), CVAI initially demonstrated the strongest association in Model 1 (β=−4.71, 95% CI: −6.17 to −3.25, p<0.001), followed by METS_VF (β=−3.68, 95% CI: −5.16 to −2.21, p<0.001). However, all female associations became non-significant after full confounder adjustment (all adjusted p>0.05).

## Discussion

This large-scale cross-sectional study, involving 1,335 patients with type 2 diabetes, aimed to investigate the associations between various anthropometric and metabolic indices and estimated glomerular filtration rate (eGFR) through

**Fig 2. Standardized Coefficients for the Association Between eGFR and Anthropometric Adiposity Indices in a Diabetic Population.** This figure illustrates the unadjusted and adjusted standardized coefficients (β) for the association between estimated glomerular filtration rate (eGFR) and various anthropometric adiposity indices in a type 2 diabetes mellitus (T2DM) population. The models are defined as follows: Model 1: Unadjusted for any variables. Model 2: Adjusted for age and sex. Model 3: Fully adjusted for age, sex, smoking history, alcohol consumption history, duration of diabetes, systolic blood pressure (SBP), diastolic blood pressure (DBP), fasting plasma glucose (FPG), total cholesterol (TC), high-density lipoprotein cholesterol (HDL), low-density lipoprotein cholesterol (LDL), and history of angiotensin receptor blocker (ARB) use.

comprehensive statistical analysis including correlation analysis, overall population regression modeling, and sex-stratified approaches. The comprehensive statistical analysis revealed profound sex-specific differences through the complete analytical progression from correlation to multivariate modeling. Sex-stratified correlation analysis (Fig 3) demonstrated striking disparities: female participants exhibited extensive significant correlations in unadjusted analysis, with CVAI showing remarkably strong association (r = −0.274, p < 0.001), while males showed limited significant associations with only METS_VF achieving significance (r = −0.130, p < 0.001). However, after confounder adjustment (Fig 3B), all female associations became non-significant while VAI remained the sole robust predictor in males (adjusted r = −0.103, p < 0.01). The multivariate regression analyses (Figs 4 and 5) confirmed these patterns, demonstrating VAI's independent predictive value in males (β = −2.41, 95% CI: −3.84 to −0.98, p = 0.001) while showing complete attenuation of female associations after full adjustment. This analytical progression illuminated fundamental mechanistic differences: female associations were primarily mediated through confounding pathways, while male associations, though initially weaker, proved more independent

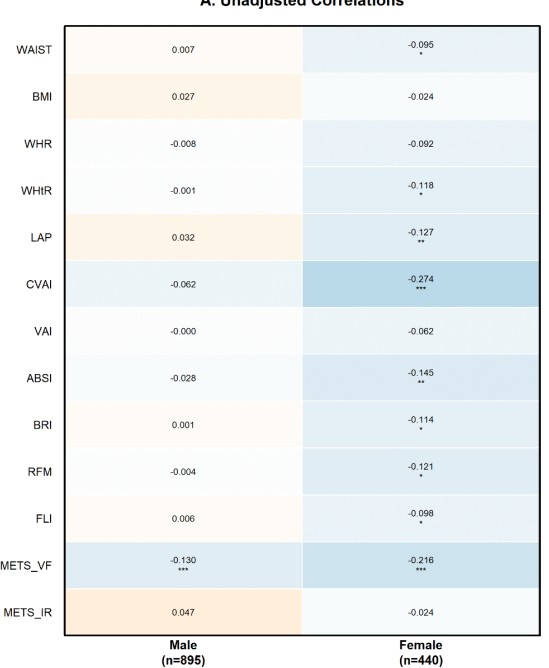

**Fig 3. Sex-stratified correlations between visceral adiposity indices and estimated glomerular filtration rate.** (A) Unadjusted Pearson correlation coefficients between 13 visceral adiposity indices and estimated glomerular filtration rate (GFR.MORD). (B) Partial correlation coefficients adjusted for potential confounding variables (AGE, SMOKING, WINE, BP, BP1, TIME, FPG, TC, HDL, LDL, ARB). Heat map colors represent the strength and direction of correlations, with blue indicating negative correlations and red indicating positive correlations. Values in cells represent correlation coefficients with statistical significance indicated by asterisks: *** P < 0.001, ** P < 0.01, * P < 0.05. Sample sizes: male participants n = 895, female participants n = 440.

and clinically relevant. These findings underscore the critical importance of comprehensive analytical approaches that progress from exploratory correlation to confirmatory regression modeling in sex-specific metabolic research.

Nevertheless, our results parallel existing literature: visceral adiposity exhibits a significant influence on the deterioration of renal function, especially among populations already facing a high metabolic burden. In a study by Kang et al. (2015), conducted on a general population of 3,185 participants, increased visceral fat area (VFA) was associated with both decreased eGFR and higher chronic kidney disease (CKD) risk (odds ratio: 1.42, 95% CI: 1.23–1.64, p < 0.001), even after adjusting for age, sex, and blood pressure [12]. Furthermore, recent meta-analyses have consistently confirmed visceral adiposity's correlation with declining kidney function in diverse populations [13,14]. Our study, focused on Chinese patients with type 2 diabetes, similarly identified that indices of visceral adiposity, such as VAI and CVAI, are important independent indicators of renal function decline.

These findings suggest that visceral adiposity may not only exacerbate systemic inflammation and oxidative stress [15–18], but also contribute to insulin resistance, endothelial dysfunction, and renin-angiotensin system activation, which collectively speed up renal damage [19,20]. Future studies, especially those including biomarker measurements or mechanistic exploration, are needed to validate these pathways among Chinese T2DM patients. There is also evidence that visceral adiposity is tightly connected to other cardiometabolic risk factors that compound its harmful impact on nephropathy [21,22]. By focusing on a high-risk group—diabetic patients with pronounced metabolic abnormalities—our study sheds light on how dysfunctional adipose tissue adversely influences renal outcomes.

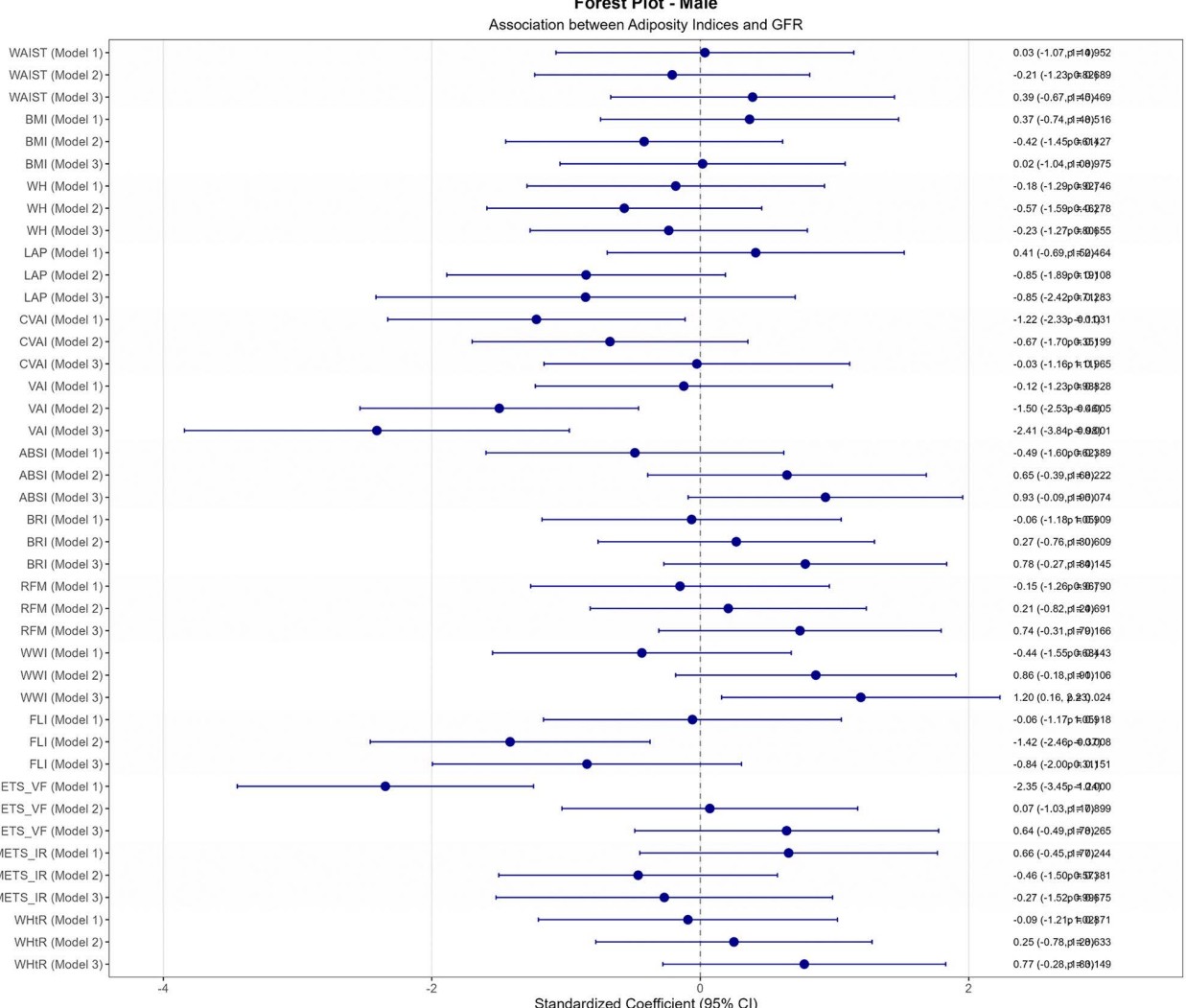

**Fig 4. Standardized Coefficient for association between eGFR and anthropometric adiposity indices in male diabetic population.** This figure illustrates the unadjusted and adjusted standardized coefficients (β) for the association between estimated glomerular filtration rate (eGFR) and various anthropometric adiposity indices in male type 2 diabetes population. The models are defined as follows: Model 1: unadjusted variables, Model 2: adjusted variables for age, Model 3: adjusted variables for age, sex, smoking history, alcohol history, duration of diabetes, SBP, DBP, FPG, TC, HDL, LDL and ARB use history.

Sex-specific differences in the association between visceral adiposity measures and renal dysfunction were evident in our findings and corroborate prior research. For instance, studies have demonstrated significant sexual dimorphism in visceral adiposity distribution and its metabolic consequences. Palmer and Clegg (2015) highlighted that men tend to accumulate more visceral adipose tissue and experience greater metabolic risks compared to women, particularly in Caucasian populations [23]. This sex-specific pattern has been further supported by Karastergiou et al. (2012), who detailed the biological basis for sex differences in adipose tissue distribution and its metabolic implications [24]. In our study, VAI showed robust predictive value in Chinese males, particularly in fully adjusted models, suggesting its utility as an effective risk stratification tool. In contrast, female-specific indices such as CVAI and METS-VF showed stronger associations in unadjusted models but lost significance after adjustment. This finding underscores the complex interplay between

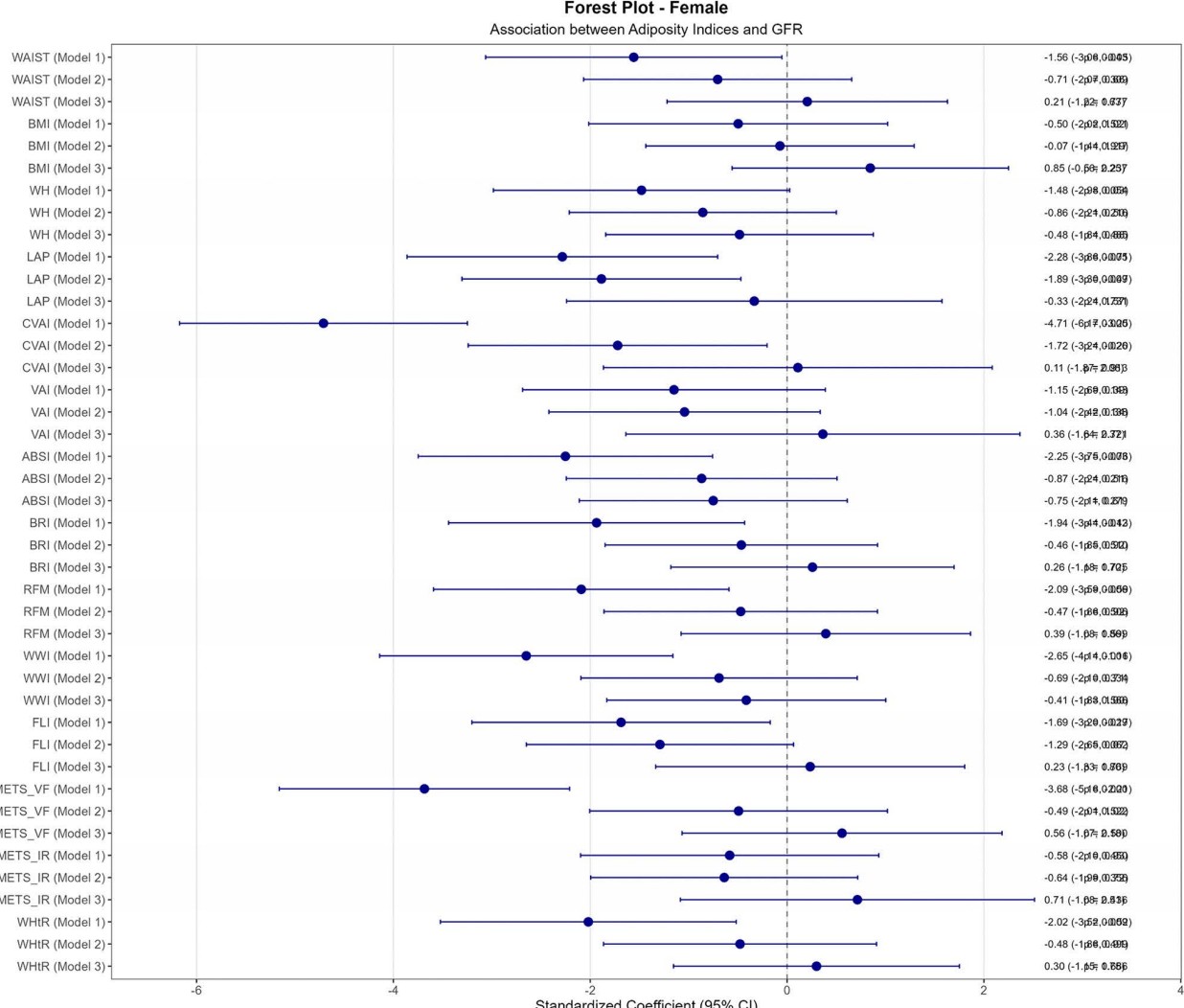

**Fig 5. Standardized Coefficient for association between eGFR and anthropometric adiposity indices in female diabetic population.** This figure illustrates the unadjusted and adjusted standardized coefficients (β) for the association between estimated glomerular filtration rate (eGFR) and various anthropometric adiposity indices in female type 2 diabetes population. The models are defined as follows: Model 1: unadjusted variables, Model 2: adjusted variables for age, Model 3: adjusted variables for age, sex, smoking history, alcohol history, duration of diabetes, SBP, DBP, FPG, TC, HDL, LDL and ARB use history.

metabolic and hormonal factors in women, possibly reflecting the protective role of estrogen on fat distribution and metabolic regulation prior to menopause, as described by White and Tchoukalova (2014) [25]. Notably, while Palmer and Clegg's study was conducted in predominantly Western populations, our results extend the understanding of sex-specific adiposity effects to Chinese patients, highlighting the context of racial and ethnic variability.

Contrasting results were observed when compared to Wang et al. (2008), who conducted a systematic review and meta-analysis examining the association between obesity and kidney disease. Their findings demonstrated that obesity-related measures were stronger predictors of kidney disease in women than in men, which is at odds with our results [26]. This discrepancy may arise from differences in study populations and methodologies. Wang et al.'s meta-analysis included a broader general population, whereas our focus was on Chinese type 2 diabetes patients—a

group characterized by a higher burden of metabolic derangements and chronic vascular inflammation, factors that may more strongly potentiate the detrimental effects of visceral adiposity in males. Methodologically, while Wang et al.'s analysis primarily utilized traditional anthropometric measures such as BMI and WHR, our study leveraged composite indices such as VAI and CVAI, which have been validated to better reflect metabolic disorders and predict renal outcomes in diabetic populations [27,28].

More recent systematic reviews have highlighted the limited utility of general measures like BMI for assessing adiposity-related metabolic risks, particularly in Asian populations [29,30]. In our study, composite indices such as VAI outperformed these simpler measures, likely because they integrate a broader range of metabolic parameters, which is particularly relevant in individuals with type 2 diabetes. These differences may highlight the advantage of composite indices in populations with high metabolic risk, like Chinese diabetes patients, who exhibit distinct adiposity profiles characterized by higher visceral fat accumulation at lower BMI levels compared to their Western counterparts [31,32]. Additionally, research suggests that even at lower BMI thresholds, Asian populations exhibit higher visceral fat content and metabolic risk compared to their Western counterparts [33,34]. These population-specific characteristics further emphasize the importance of tailored assessment approaches.

The variation in findings across studies likely reflects differences in population characteristics, adiposity metrics, and methodological approaches. Furthermore, disparities in analytical approaches and covariate adjustments may partially account for the observed differences across studies. These nuances emphasize the importance of tailoring adiposity indices and risk assessment strategies to specific populations [14,35].

Beyond methodological considerations, the observed sex-specific differences may be attributed to hormonal regulation of fat distribution, particularly the protective metabolic effects of estrogen in premenopausal women, which potentially mitigates the link between visceral adiposity and renal outcomes [36]. This mechanism helps explain why female associations became non-significant after adjustment, while VAI maintained robust predictive value in males [37,38]. The consistent male-specific associations underscore VAI's utility as a risk stratification tool in this population.

The clinical significance of our study lies in its ability to refine risk stratification for renal dysfunction in Chinese patients with type 2 diabetes through the application of visceral adiposity indices. By demonstrating the predictive value of composite indices such as VAI and CVAI, our findings advocate for a more nuanced and gender-specific approach in clinical practice. Unlike traditional indicators like BMI or WHR, these indices integrate metabolic and anthropometric measurements to provide more accurate renal risk assessment. Given their simplicity and accessibility, they have potential as pragmatic tools in resource-limited settings for earlier identification of high-risk individuals, particularly male patients who showed consistent VAI-eGFR associations. Policymakers could consider integrating these indices into CKD screening guidelines for diabetic populations to optimize resource allocation and prioritize preventive interventions.

This study has several limitations to consider. First, as a single-center study on Chinese type 2 diabetes patients, the generalizability of our findings to other populations or settings is limited, and caution is needed when applying these results to different ethnic groups or demographics. Second, the exclusion of patients with severe chronic conditions may limit the applicability of our findings to populations with complex comorbidities. Third, as an observational study, causality between visceral adiposity and renal dysfunction cannot be established, and residual confounding or reverse causation may influence the results. Fourth, while we adjusted for measurable confounders such as glycemic control and blood pressure, unmeasured factors, including genetic predispositions or lifestyle behaviors, could have impacted the findings. Fifth, the inherent nature of retrospective data collection led to some incompleteness in records, although this was mitigated by rigorous multiple imputation methods. Lastly, the reliance on anthropometric indices rather than advanced imaging techniques may introduce some inaccuracy in assessing visceral fat. Future multicenter, longitudinal studies in diverse populations using imaging techniques are needed to validate these findings and explore causality. These limitations underscore the importance of cautious interpretation while guiding future research in metabolic and renal health.

Future research should focus on prospective multicenter studies to validate the prognostic utility of VAI and CVAI across diverse ethnic groups, assessing their predictive value for hard endpoints such as CKD progression, dialysis initiation, and cardiovascular events. Additionally, investigating targeted interventions aimed at reducing visceral adiposity—including lifestyle modifications and pharmacological strategies—could provide actionable pathways to improve metabolic health and protect renal function. Exploring the biological mechanisms underpinning visceral adiposity-renal dysfunction relationships, particularly the roles of systemic inflammation, insulin resistance, and oxidative stress, will be crucial for developing comprehensive treatment approaches.

In summary, our comprehensive analysis demonstrates significant sex-specific associations between visceral adiposity indices and renal dysfunction in Chinese T2DM patients, with VAI showing particular utility in male risk stratification. These findings support the integration of composite adiposity indices into routine clinical assessment and highlight the necessity of population-specific, gender-tailored approaches in diabetic nephropathy prevention. Longitudinal studies are needed to establish causality and evaluate targeted interventional strategies.

## Supporting information

**S1 Dataset. De-identified minimal underlying data.** This file contains the raw data for all 1,335 participants included in the final analysis, used to generate the findings presented in the study.
(XLSX)

## Author contributions

**Data curation:** Songfang Liu, Louyan Ma, Ranran Ma, Ting Qi.

**Formal analysis:** Louyan Ma, Yu Niu.

**Investigation:** Ranran Ma, Ting Qi.

**Methodology:** Songfang Liu, Louyan Ma, Ranran Ma, Ting Qi.

**Project administration:** Bingyin Shi.

**Supervision:** Bingyin Shi.

**Writing – original draft:** Songfang Liu.

**Writing – review & editing:** Yu Niu, Bingyin Shi.

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
