## [Decision Letter · Decision Letter 0]

29 Dec 2025

Dear Dr. Shi,

Thank you for submitting your manuscript to PLOS ONE. After careful consideration, we feel that it has merit but does not fully meet PLOS ONE’s publication criteria as it currently stands. Therefore, we invite you to submit a revised version of the manuscript that addresses the points raised during the review process.

We look forward to receiving your revised manuscript.

Kind regards,

Aleksandra Klisic

Academic Editor

PLOS One

Journal Requirements:

4. In the online submission form, you indicated that the datasets generated during and/or analysed during the current study are not publicly available due to the inclusion of detailed experimental parameters and raw data from ongoing related research projects, but are available from the corresponding author on reasonable request.

Reviewers' comments:

Reviewer's Responses to Questions

**Comments to the Author**

1. Is the manuscript technically sound, and do the data support the conclusions?

Reviewer #1: Yes

Reviewer #2: Yes

2. Has the statistical analysis been performed appropriately and rigorously?

Reviewer #1: Yes

Reviewer #2: Yes

3. Have the authors made all data underlying the findings in their manuscript fully available?

Reviewer #1: Yes

Reviewer #2: Yes

4. Is the manuscript presented in an intelligible fashion and written in standard English?

Reviewer #1: Yes

Reviewer #2: Yes

Reviewer #1: Thank you dear authors for conducting such important public health issue; saying this I would like to give my comments and I need some clarifications.

1. As much as possible it is better to avoid abbreviations on the Abstract section

2. The Abstract section lacks background which summarizes the magnitude and severity of renal dysfunction among T2DM patients and the real gaps filled by this specific study

3. How did you calculate the sample size in accordance with your source population manually or using software? 1300 Vs 1335 is unclear.

4. What is your outcome variable? it hasn't been well emphasized

5. What were the limitations of your study?

6. How did you resolve incompleteness of records, eligibility issues for the incompletes

7. What was the main reason why you didn't you conduct a community based study and collect primary data?

Lastly, it is better to present your findings using tables and figures as well.

Reviewer #2: Its a really interesting topic, it must have been well thought before execution began. The research also has a generous sample size which makes it very valid. Your results also give a positive correlation to your topic, which is quite good.

A few things i would like to point out include

1. Your figures should have been in the body of the results and not the appendix to make the work easier to understand. While the tables are okay in the appendix, the figures should be in the body of your work.

2. Instead of the meaning of all the abbreviations for indices being under table 2 in the appendix. It should preferably be in a list form and very bold so it can be properly seen. preferably just after the disclosure statement.

**Do you want your identity to be public for this peer review?** For information about this choice, including consent withdrawal, please see our Privacy Policy

Reviewer #1: No

Reviewer #2: **Yes:** ADEFUSI TEMILOLUWA

---

## [Author Response · Author response to Decision Letter 1]

7 Jan 2026

Response to Journal Requirements:

1.Formatting and Style:

We have reviewed the PLOS ONE style requirements and adjusted the file naming and manuscript formatting accordingly.

2.ORCID iD:

The corresponding author has updated their information in Editorial Manager and validated their ORCID iD as requested.

3.Ethics Statement:

We confirm that the ethics statement is now exclusively located in the Methods section of the manuscript.

4.Data Availability:

We have updated our Data Availability Statement. The minimal data set underlying the findings described in our manuscript has been uploaded as Supporting Information (S1 Dataset).

5. References:

We have checked the reference list for completeness and correctness, ensuring no retracted papers are cited.

Response to Reviewer #1

1.As much as possible it is better to avoid abbreviations on the Abstract section.

Reply:

Thank you for this suggestion. We have revised the Abstract to minimize abbreviations. Terms such as T2DM, eGFR, and VAI have been spelled out where appropriate to improve readability.

Changes made: [See Abstract, Page 1-2]

2.The Abstract section lacks background which summarizes the magnitude and severity of renal dysfunction among T2DM patients and the real gaps filled by this specific study.

Reply:

We have added a background sentence to the Abstract to contextualize the study. It now briefly highlights the impact of renal dysfunction in diabetes and the gap regarding visceral adiposity indices in the Chinese population.

Changes made: [See Abstract, Page 1]

3.How did you calculate the sample size in accordance with your source population manually or using software? 1300 Vs 1335 is unclear.

Reply:

We apologize for the confusion. The sample size of 1,300 was our calculated target to ensure sufficient power for sex-stratified analysis. The final number of eligible participants enrolled was 1,335, which met and slightly exceeded this target. We have clarified this distinction in the "Sample Size Calculation" section.

Changes made: [See Methods, "Sample Size Calculation" section, Page 5]

4.What is your outcome variable? it hasn't been well emphasized.

Reply:

We have revised the text to explicitly state that the primary outcome variable is renal dysfunction, defined as an eGFR < 60 mL/min/1.73 m².

Changes made: [See Methods, "Variables" section, Page 6-7]

5.What were the limitations of your study?

Reply:

We have reviewed the "limitations" paragraph in the Discussion section. We explicitly acknowledge the single-center design, the cross-sectional nature preventing causal inference, and potential residual confounding. We have also added a note regarding the retrospective nature of data collection.

Changes made: [See Discussion, Page 21]

6.How did you resolve incompleteness of records, eligibility issues for the incompletes.

Reply:

Participants with missing data on core variables (key visceral fat markers or renal function) were excluded from the study as per our exclusion criteria. For covariates with minimal missing data (<5%), we utilized multiple imputation to minimize bias and maximize data utility. We have clarified this in the Statistical Analysis section.

Changes made: [See Methods, "Statistical Analysis" section, Page 7-8]

7.What was the main reason why you didn't you conduct a community based study and collect primary data?

Reply:

This is a valuable question. We chose a hospital-based retrospective design for two main reasons:

Clinical Precision: Hospital records provide more comprehensive and precise laboratory measurements (e.g., specific biochemical markers) compared to what is typically feasible in large-scale community screenings.

Efficiency: Utilizing electronic medical records allowed us to efficiently analyze a large cohort of confirmed T2DM patients to generate hypotheses regarding sex-specific differences, which can subsequently be tested in future prospective community-based studies.

8. Lastly, it is better to present your findings using tables and figures as well.

Reply:

Thank you for this suggestion. We have ensured that our findings are comprehensively presented using Tables and Figures. Regarding the placement of figures, we initially moved them into the text body per your suggestion; however, in strict adherence to the journal’s submission guidelines received from the Editorial Office, we are required to upload figures as separate high-resolution files (TIFF/EPS) rather than embedding them in the manuscript. We have ensured that the Figure Legends are clearly placed within the Results section to indicate the logical flow of the findings.

Changes made: [Figures have been uploaded as separate files per journal policy; Figure Legends are retained in the Results section]

Response to Reviewer #2

1.Your figures should have been in the body of the results and not the appendix to make the work easier to understand. While the tables are okay in the appendix, the figures should be in the body of your work.

Reply:

We fully appreciate and agree with the reviewer’s view that placing figures within the text improves readability. However, we have received specific technical instructions from the Editorial Office requiring us to remove embedded images from the manuscript file and upload them as separate standard image files (TIFF/EPS). Therefore, to comply with the journal’s production requirements, we have uploaded the figures separately. The submission system will automatically compile these files into the final PDF for your review. We have placed the Figure Legends within the body of the Results section to assist with the reading flow.

Changes made: [Embedded figures were removed and uploaded as separate files to comply with editorial instructions]

2.Instead of the meaning of all the abbreviations for indices being under table 2 in the appendix. It should preferably be in a list form and very bold so it can be properly seen. preferably just after the disclosure statement.

Reply:

We have created a bolded List of Abbreviations and placed it immediately after the Disclosure statement as requested, ensuring all indices are clearly defined for the reader.

Changes made: [See Page 23]

---

## [Decision Letter · Decision Letter 1]

21 Jan 2026

Dear Dr. Shi,

Thank you for submitting your manuscript to PLOS ONE. After careful consideration, we feel that it has merit but does not fully meet PLOS ONE’s publication criteria as it currently stands. Therefore, we invite you to submit a revised version of the manuscript that addresses the points raised during the review process.

We look forward to receiving your revised manuscript.

Kind regards,

Aleksandra Klisic

Academic Editor

PLOS One

Journal Requirements:

Reviewers' comments:

Reviewer's Responses to Questions

**Comments to the Author**

Reviewer #2: (No Response)

2. Is the manuscript technically sound, and do the data support the conclusions?

Reviewer #2: Yes

3. Has the statistical analysis been performed appropriately and rigorously?

Reviewer #2: Yes

4. Have the authors made all data underlying the findings in their manuscript fully available?

Reviewer #2: Yes

5. Is the manuscript presented in an intelligible fashion and written in standard English?

Reviewer #2: Yes

Reviewer #2: The article has been properly revised, it is concise and precise to read. I appreciate that the tables have been incorporated in the results section for easy correlation too and plagiarism issues have been addressed.

However, i still do not think it right that we have picture legends and footnotes in the midst of the article and the pictures aren't preceding it. it makes more sense that the picture legends and footnotes are where the pictures are, either in the body of work or in the appendix. Thank you.

**Do you want your identity to be public for this peer review?** For information about this choice, including consent withdrawal, please see our Privacy Policy

Reviewer #2: **Yes:** ADEFUSI TEMILOLUWA

---

## [Author Response · Author response to Decision Letter 2]

25 Jan 2026

Response to Reviewer #2

We sincerely thank the reviewer for this thoughtful comment and fully understand the concern regarding the placement of figure legends and footnotes.

However, we would like to clarify that this format is in accordance with PLOS ONE's submission requirements. During our previous revision submission, when we included figures alongside their legends in the main text (following the reviewer's preference), the submission failed the journal's format check. We received the following specific instruction from the journal: "Please remove your figures from within your manuscript file, leaving only the individual TIFF/EPS image files. These will be automatically included in the reviewer's PDF."

Following this guidance, we removed all embedded figures and uploaded them as separate image files. Therefore, the current format—with figure legends in the manuscript and figures as separate files—is in compliance with PLOS ONE's submission guidelines. Once the manuscript is accepted and published, the figures and their corresponding legends will be properly integrated in the final published version.

We sincerely appreciate the reviewer's understanding on this matter.

---

## [Decision Letter · Decision Letter 2]

5 Feb 2026

Sex-specific Association of Visceral Adiposity Index with Renal Dysfunction in Chinese Type 2 Diabetes: A Cross-sectional Study

PONE-D-25-64102R2

Dear Dr. Shi,

We’re pleased to inform you that your manuscript has been judged scientifically suitable for publication and will be formally accepted for publication once it meets all outstanding technical requirements.

Kind regards,

Aleksandra Klisic

Academic Editor

PLOS One

Additional Editor Comments (optional):

Reviewers' comments:

Reviewer's Responses to Questions

**Comments to the Author**

Reviewer #2: All comments have been addressed

2. Is the manuscript technically sound, and do the data support the conclusions?

Reviewer #2: Yes

3. Has the statistical analysis been performed appropriately and rigorously?

Reviewer #2: Yes

4. Have the authors made all data underlying the findings in their manuscript fully available?

Reviewer #2: Yes

5. Is the manuscript presented in an intelligible fashion and written in standard English?

Reviewer #2: Yes

Reviewer #2: All necessary comments raised have been addressed. It is a very insightful and interesting study. Well done

**Do you want your identity to be public for this peer review?** For information about this choice, including consent withdrawal, please see our Privacy Policy

Reviewer #2: **Yes:** Adefusi Temiloluwa

---

## [Editor Report · Acceptance letter]

PONE-D-25-64102R2

PLOS One

Dear Dr. Shi,

I'm pleased to inform you that your manuscript has been deemed suitable for publication in PLOS One. Congratulations! Your manuscript is now being handed over to our production team.

Kind regards,

on behalf of

Dr. Aleksandra Klisic

Academic Editor

PLOS One